# Extended Prophylactic Effect of N-Tert-Butyl-α-Phenylnitron against Oxidative/Nitrosative Damage Caused by the DNA-Hypomethylating Drug 5-Azacytidine in the Rat Placenta

**DOI:** 10.3390/ijms23020603

**Published:** 2022-01-06

**Authors:** Nikola Sobočan, Marta Himelreich-Perić, Ana Katušić-Bojanac, Jure Krasić, Nino Sinčić, Željka Majić, Gordana Jurić-Lekić, Ljiljana Šerman, Andreja Marić, Davor Ježek, Floriana Bulić-Jakuš

**Affiliations:** 1Scientific Centre of Excellence for Reproductive and Regenerative Medicine, School of Medicine, University of Zagreb, 10000 Zagreb, Croatia; sobocann@gmail.com (N.S.); ana.katusic@mef.hr (A.K.-B.); jure.krasic@mef.hr (J.K.); nino.sincic@mef.hr (N.S.); gjuric@mef.hr (G.J.-L.); sermanl@mef.hr (L.Š.); davor.jezek@mef.hr (D.J.); floriana@mef.hr (F.B.-J.); 2Department of Gastroenterology, University Hospital Merkur, 10000 Zagreb, Croatia; 3Department of Biology, School of Medicine, University of Zagreb, 10000 Zagreb, Croatia; majiczeljka@gmail.com; 4Department of Histology and Embryology, School of Medicine, University of Zagreb, 10000 Zagreb, Croatia; 5Department of Internal Medicine, County Hospital Čakovec, 40000 Čakovec, Croatia; anmar44@gmail.com

**Keywords:** 5-azacytidine, N-tert-Butyl-α-phenylnitron, PBN, placenta, oxoguanosine, nitrotyrosine, oxidative stress

## Abstract

Antioxidant N-tert-Butyl-α-phenylnitron (PBN) partly protected embryos from the negative effects of a DNA demethylating drug 5-azacytidine during pregnancy. Our aim was to investigate PBN’s impact on the placenta. Fischer rat dams were treated on gestation days (GD) 12 and 13 by PBN (40 mg/kg), followed by 5azaC (5 mg/kg) after one hour. Global methylation was assessed by pyrosequencing. Numerical density was calculated from immunohistochemical expression in single cells for proliferating (PCNA), oxidative (oxoguanosine) and nitrosative (nitrotyrosine) activity. Results were compared with the PBN-treated and control rats. PBN-pretreatment significantly increased placental weight at GD15 and GD20, diminished by 5azaC, and diminished apoptosis in GD 20 placentas caused by 5azaC. Oxoguanosine expression in placentas of 5azaC-treated dams was especially high in the placental labyrinth on GD 15, while PBN-pretreatment lowered its expression on GD 15 and GD 20 in both the labyrinth and basal layer. 5azaC enhanced nitrotyrosine level in the labyrinth of both gestational stages, while PBN-pretreatment lowered it. We conclude that PBN exerted its prophylactic activity against DNA hypomethylating agent 5azaC in the placenta through free radical scavenging, especially in the labyrinthine part of the placenta until the last day of pregnancy.

## 1. Introduction

During placental development, the level of oxygen and free radical production is changing from low to high, being low during the first trimester of human development, intermediate during the second trimester when the uteroplacental circulation is established, and finally high in the third trimester when the placenta is fully developed to sustain increased growth of the fetus. This is accompanied by changes in antioxidant production from low to increased [1]. The imbalance between free radical generation and antioxidant production leads to adverse pregnancy outcomes based on lipids, proteins, and DNA damage in the placental tissue and its accelerated aging [2,3]. Various antioxidants have been studied with the aim to prevent that imbalance and pregnancy-related pathologies [4,5,6].

Nitrones, among which N-tert-Butyl-α-phenylnitron (PBN) is a member, act in various pathways that change the redox state through scavenging reactive oxygen species (ROS) and/or reactive nitrogen species (RNS). They release NO and control gene expression [7,8]. Nitrones, being the spin-traps, intercept ROS before any damage is done and do not interact with the physiological oxygen [9]. As the best-known role of nitrones is ROS scavenging, they are often classified as antioxidants. The spin-trap PBN [10] protects mammalian embryos from the teratogenic activity of phenytoin, benzo[a]pyrene, thalidomide, ethanol and methanol which shows that their teratogenic activity involves reactive oxygen species [11,12]. The antioxidative effect of acetylsalicylic acid (ASA) inhibits phenytoin-initiated oxidative damage to protein and lipids in embryonic tissues [13]. We showed that ASA was also able to ameliorate the negative impact on rat embryos and placentas subjected to the epigenetic drug 5-azacytidine (5azaC) [14,15], which is used for the treatment of human diseases such as myelodysplastic syndrome [16]. 5azaC was called a “genomic medicine” because it targets the whole genome by blocking DNA-methyltransferase in its activity [17]. 5azaC converts to deoxy-azacytidine (DAC) that is incorporated into DNA (10–20%) exhibiting a DNA-hypomethylating activity at numerous targets such as promoters of tumor suppressor genes, oncogene gene bodies, endogenous retroviral elements, micro RNA/long non-coding RNAs promoters. DAC is usually leading to impaired proliferation and apoptosis. On the other hand, 5azaC massively binds to newly synthesized RNA (80–90%) leading to mRNA and protein metabolism disruption [18,19].

DNA methylation is an epigenetic mechanism with specific dynamics during normal mammalian development and aging of the adult organism [20]. During placental development, DNA hypomethylating agent 5azaC may be detrimental to the placenta [21,22,23] because DNA methylation is involved in the regulation of gene activity necessary for differentiation and other developmental processes [24,25,26]. Placental methylome characteristics in various environmental exposures and pregnancy disorders have recently been reviewed as representing the focal interest in epigenetic investigation of the placenta. However, the complete insight into the drivers and passengers involved still represents a challenge, especially because epigenetic mechanisms, such as DNA methylation, histone posttranscriptional modifications, and RNAi are known to collaborate in epigenetic regulation of gene expression [18,27,28].

The association of DNA methylation and ROS has been extensively studied especially in cancers where ROS can influence both DNA hypermethylation and hypomethylation [29]. We have recently shown that the activity of the DNA hypomethylating agent 5azaC, which caused intrauterine growth restriction (IUGR) and induced specific malformations in rat fetuses, was ameliorated by the free-radical scavenger PBN [30]. IUGR is usually associated with placental insufficiency [31] and our previous results showed that 5azaC impaired placental development and changed its glycoprotein composition [21,22,23]. As recently reviewed, placental weight decrease leading to IUGR is mainly a result of apoptosis or necrosis of trophoblast cells and hypoplasia of the labyrinth zone and to fully understand the developmental toxicity of various agents, they should be investigated in a stage-specific and tissue-specific manner [32,33].

We now hypothesize that the negative activity of the DNA hypomethylating drug 5azaC on the developing rat placenta is also associated with free radicals, which could be diminished by the pretreatment with a free-radical scavenger. For our experiments, we have chosen treatment on days 12–13 of gestation (GD 12–13), characterized by the endovascular trophoblast invasion and intense proliferation of the labyrinth and basal zone, while the results were evaluated on GD 15 and on a day before birth (GD 20) when the labyrinth zone is maximally developed [32,34]. Our experiments confirmed the association of free radicals, elicited by 5azaC-treatment during the critical phase of placental development with apoptosis, as well as alleviation of this effect by the PBN-pretreatment.

## 2. Results

### 2.1. PBN-Pretreatment Increases the Placental Weight in 5azaC Treated Dams

We first weighted the placentas isolated from a large group of animals to confirm the negative effect of the 5azaC treatment and assess the possible ameliorating effect of PBN pretreatment in rat dams [21]. For GD 15 we measured placental weights from 33 controls, 35 treated with 5azaC, 57 treated with PBN, 52 treated with PBN/azaC and for GD 20 from 32 controls, 36 treated with 5azaC, 56 treated with PBN and 51 treated with PBN/5azaC (Figure 1).

As expected, the weight of placentas from all groups of dams increased from day 15 (Figure 1A) to day 20 of gestation (Figure 1B). PBN alone did not interfere with the growth of placentas in comparison to controls neither in GD 15 (Figure 1A) nor GD 20 (Figure 1B). Compared to control and PBN-treated groups, 5azaC-treatment of dams significantly diminished the placental weight in both gestational stages, i.e., in GD 15 (Figure 1A) (*p* < 0.0001) and GD 20 (Figure 1B) (*p* < 0.0001). PBN-pretreatment significantly increased the placental weight of 5azaC-treated dams, more significantly on GD 15 (Figure 1A) (*p* < 0.001) than on GD 20 (Figure 1B) (*p* < 0.05). PBN-pretreatment did not rescue placental growth to the level found in controls because the placental weight was significantly lower than in controls and PBN-treated dams in GD 15 (Figure 1 A) (*p* < 0.0001) and GD 20 (Figure 1B) (*p* < 0.0001). Both in GD 15 and GD 20, the weight of placentas treated with PBN only compared to PBN pretreatment with 5azaC was significantly increased (*p* < 0.0001).

These results confirmed that the DNA hypomethylating drug 5azaC applied to pregnant dams at the critical phase of placental development (GD 12–13) diminished overall placental growth and that this effect could be ameliorated by the pretreatment with PBN, which can thus serve as a prophylactic agent.

### 2.2. Global DNA Methylation in Placentas

The level of global DNA methylation in placentas was assessed in 6 samples from each group of treated pregnant dams (Figure 1C). In placentas of dams treated with the DNA-hypomethylating drug 5azaC global DNA methylation was significantly lower (32–35%) (*p* < 0.001) than in dams treated only with PBN (35–38%), but not in comparison to controls where the percentage of methylation varied the most between individual samples (31–38%). PBN used as pretreatment did not enhance global DNA methylation levels (31–35%) (Figure 1C).

### 2.3. PBN and Cell Proliferation in Placentas of 5azaC-Treated Dams

Since we have shown that the effect of the PBN-pretreatment ameliorates the negative effect of 5azaC treatment on placental weight, we investigated whether this effect was due to the PBN positive impact on the cell proliferation. The potential for cell proliferation was assessed by quantifying proliferating cell nuclear antigen (PCNA) expression (Figure 2A–D) with the stereological parameter numerical density (Nv). That analysis was possible only in GD 20 placentas because of the rarity of PCNA signals in very small placentas of GD 15. Our results (Figure 2) showed that PBN alone did not cause significant aberration from the control values. Similar, significantly lower Nv for the PCNA in placentas of both groups of dams treated with 5azaC compared to controls or PBN-treated was found (*p* < 0.0001). We conclude that PBN-pretreatment does not ameliorate placental weight at preterm through cell proliferation.

### 2.4. PBN Diminishes Apoptosis in Placentas of 5azaC Treated Dams

In search of the cellular mechanisms by which the PBN-pretreatment enhances placental weight, we assessed the apoptotic index in treated dams’ placentas (Figure 3). On GD 15, no differences were found among groups. However, on GD 20, 5azaC-treatment significantly enhanced apoptosis in placentas in comparison to controls and PBN-treated rats (Figure 3) (*p* < 0.0001). Importantly, PBN-pretreatment significantly lowered the level of apoptosis in placentas of 5azaC-treated dams (*p* < 0.0001). Therefore, the prophylactic activity of PBN is associated with the prevention of apoptosis at GD 20.

### 2.5. PBN-Pretreatment Diminishes Oxidative Stress Induced by 5azaC in Placentas of Treated Dams

In placentas of treated dams, we have pinpointed apoptosis as the cellular mechanism that was prevented by the ROS/RNS scavenger PBN, for as long as six days after the treatment with 5azaC, namely until GD 20. We wanted to compare the levels of ROS/RNS markers soon after the treatment with 5azaC on GD 15 and then on GD 20. This was done in two different placental compartments, the labyrinth and the basal layer, because it is known that the labyrinth is more affected by 5azaC treatment than the basal layer [22,32].

#### 2.5.1. 8-Hydroxy-2′deoxyguanosine (8-OHdG)

Oxidative stress was first investigated in single-placenta cells using an oxoguanosine (8-OHdG) marker in placental samples of all three groups of treated animals and in controls (Figure 4).

8-OHG positivity was found mainly in the cytoplasm. Although we also observed 8-OHdG positivity in some nuclei, this was too scarce to quantify. Quantification of cytoplasmatic signals showed that the 8-OHdG was expressed mainly in the labyrinth of GD 15 placentas, where the highest level was induced by 5azaC (Figure 4B) (*p* < 0.0001). Still, pretreatment with PBN significantly lowered 8-OHdG level, with a different intensity in the labyrinth and the basal layer (*p* < 0.0001 and *p* < 0.001, respectively), but not to the control levels (*p* < 0.0001). Unexpectedly, in the labyrinth, PBN alone significantly raised the level of 8-OHdG in comparison with controls (*p* < 0.0001).

In GD 20 placentas, 8-OHdG was absent from both the labyrinth and the basal layer of PBN-treated dams (Figure 4E,F). At the same time, PBN-pretreatment significantly lowered the level of 8-OHdG in placentas of 5azaC-treated dams (*p* < 0.0001), although not to control levels (*p* < 0.001).

#### 2.5.2. Nitrotyrosine

Oxidative stress was investigated using a nitrotyrosine (NT) marker in placental samples of all three groups of treated animals and in controls (Figure 5).

The cytoplasmic expression of NT was the highest in labyrinths of GD 15 and GD 20 placentas (Figure 5B,E) of dams treated with 5azaC, while PBN-pretreatment lowered it significantly (*p* < 0.0001). In comparison to controls, PBN alone significantly increased NT level in GD 15 labyrinths (Figure 5B) (*p* < 0.001) and significantly lowered it in GD 20 labyrinths (Figure 5E) (*p* < 0.001). In the basal layer, the level of NT was much lower in GD 15 placentas than in the labyrinth, while in GD 20 placentas, its expression was insufficient for quantification, as well as in the GD 15 basal layer control samples. It may be concluded that nitrosative stress marker was still high in labyrinths after six days of treatment with 5azaC but diminished significantly by the prophylactic activity of PBN.

## 3. Discussion

Our experimental results have shown that prophylactic in vivo pretreatment of the pregnant rat dams with the antioxidant PBN generally alleviated induction of ROS/RNS by a DNA hypomethylating drug in the mammalian placenta. The only exception was in the basal layer of GD 15 placentas where pretreatment caused a significantly higher level of nitrotyrosine than in 5azaC-treated samples. The treatment with PBN only was mostly associated with lower levels of ROS/RNS markers than in controls. However, in the GD 15 labyrinth PBN-treatment alone was correlated with significantly higher levels of both oxoguanosine and nitrotyrosine than in controls. All of the above unexpected findings are possibly associated with other PBN activities than antioxidative, such as the activity of a nitrone on gene expression [35,36]. We also found that at GD 15 nitrotyrosine in the basal layer of the placenta was of a low, but detectable level in all treated except in controls, while at GD 20 it was always too low for quantification. Nitrosative stress was still high in labyrinths after six days of treatment with 5azaC but diminished significantly by the prophylactic activity of PBN. It was described before that nitrotyrosine residues were present in the placenta in association with altered placental function caused by maternal diseases [37] that obviously influenced mothers more than our dams treated with 5azaC. Importantly, protein nitrosylation in animals seems to be crucial for the function of histone deacetylases (HDACs) and histone acetyltransferases (HATs) that are involved in epigenetic regulation of gene transcription together with DNA methylation [38,39,40,41].

DNA methylation is an epigenetic mechanism with specific dynamics during normal mammalian development [42] and aging of the adult organism, where age-specific drift in DNA methylation is in the global hypomethylation and local hypermethylation [43]. While DNA methylation is mainly described as a regulator of gene expression, ROS/RNS influenced signal transduction, known as “redox signaling”, is also important for development and aging [44,45]. In our global DNA methylation research, we used the specific rat B1 ID element/SINE [46]. Such SINEs are appropriate for the analysis of global DNA methylation in the rat because they are present in over one hundred thousand copies per haploid genome [47,48]. However, in comparison to controls and treatment with only PBN, PBN pretreatment did not enhance global DNA methylation in whole 20 GD placentas as we previously found in limb buds. Results from a recent placenta study in normal pregnancies, dealing with genome-wide methylation, showed variability of methylation among samples from different pregnancies in contrast to cord blood samples. Moreover, LINE1 hypomethylation was more pronounced in placentas of smaller children stratified by birth weight percentiles [49]. Both results are in concordance with our findings, such as the more pronounced variability of methylation in control placentas [Figure 1C] and IUGR found in offspring of 5azaC treated rat dams [30].

The association of DNA methylation and ROS has been studied extensively in cancer. ROS is presumed to be the cause of both regional CpG island hypermethylation in tumor suppressor genes and generalized genomic hypomethylation. 8-hydroxy-2′-deoxyguanosine (8-OHdG), that inhibits DNA methylation at nearby cytosine bases, has been proposed as the probable cause of DNA demethylation [29]. Our results obtained on inbred animals in comparison to untreated controls have shown that a known DNA hypomethylating agent is associated to elevated levels of ROS markers, specifically 8-hydroxy-2′-deoxyguanosine (8-OHdG) and nitrotyrosine in the placenta. Therefore, it is possible that DNA demethylation *per se* can induce oxidative stress that damages both DNA and proteins and not only *vice versa.* However, one must not forget that 5azaC is able to interact directly at the RNA-level with protein production [18,19] that may be addressed in some future experiments.

Although the 8-OHdG expression is generally known to be found in nuclei [30], in the present research, 8-OHdG was expressed mainly in the cytoplasm of the placental cells. This finding is in concordance with reports of 8-OHdG accumulation in mitochondrial DNA, resulting in cell death; this again emphasizes the importance of analyzing 8-oHdG cytoplasmic signal [50,51,52,53].

Results from this study, on 5azaC-DNA hypomethylating drug induction of oxidative stress and its prophylaxis by PBN, are in concordance with the results obtained in the developing rat embryo and fetus, where we have shown that PBN-pretreatment, in addition to lowering the levels of ROS/RNS induced by 5azaC, alleviates IUGR and severity of malformations [30]. Our results confirmed that the DNA hypomethylating drug 5azaC applied to pregnant dams at the critical phase of placental development (GD 12–13) diminished overall placental growth and that this effect could be ameliorated by the pretreatment with PBN, which can thus serve as a prophylactic agent.

IUGR caused by an extraneous toxic substance is usually, but not always, associated with specific changes in placental weight that may be lower, as in the case of 5azaC, or even higher (hypertrophy) than normal, e.g., in indomethacin or alcohol exposure (for review, see [54]). The major pathohistological effect of 5-azacytidine in previous studies was on the hypoplasia of the labyrinth zone. Such an effect can be associated with the critical period of labyrinth development we have chosen for the treatment (GD 12–13) and the higher level of apoptosis that we detected. Interestingly, two days after the treatments, we found no apoptotic changes regardless of the treatment but the changes were significant at the end of the pregnancy, when placental apoptosis represents a part of normal placental aging [33]. Notably, at this late pre-term stage, the prophylactic effect of PBN was significant, which leads to the conclusion that such prophylaxis has a long-term impact. Indeed, a recent meta-analysis and meta-regression leads to the conclusion that low-dose aspirin prophylaxis applied before the 16th week of human pregnancy should result in a reduction of adverse perinatal outcomes such as perinatal death [6]. Ogbodo et al. proposed testing of free radicals and antioxidant status in early pregnancy and even before conception [55]. It is possible that future basic research using antioxidants before pregnancy or its earlier stages than GD 12–13, done in this and our previous research [30] may further support that proposition.

Although there are differences between the rat and the human placentas, they are both discoid and hemochorial. Our original experimental data obtained in the rat model, based on the ROS/RNS investigation in different tissues of the placenta, may serve to better understand ROS/RNS effects in human placentas [32].

Regarding cell proliferation, 5azaC-treatment diminished it and PBN prophylaxis had no influence in the GD 20 placentas and does not ameliorate placental weight at preterm through cell proliferation. Our previous results, obtained in the embryos of rat dams subjected to the same 5azaC treatment, showed impaired cell proliferation in limb buds but enhanced proliferation in the vertebral cartilage. In the embryonic liver, PBN pretreatment even normalized proliferation that had been diminished significantly by 5azaC [30]. Therefore, when it comes to their influence on cell proliferation, 5azaC and PBN-pretreatment exert tissue specificity, which may be caused by differential methylation levels in tissues [56] and also by varying ROS/RNS cellular levels and even their interactions in close subcellular localizations [57,58].

We conclude that in the placenta of rat dams treated with a DNA-hypomethylating drug 5azaC, the free-radical scavenger PBN exerted a prolonged prophylactic activity through both ROS and RNS free radical scavenging, especially in the labyrinthine part of the placenta. These results may be important for the prevention of premature placental aging associated with oxidative stress that leads to adverse pregnancy outcomes [59]. Additionally, our results may help understand environmental toxicity caused by DNA methylation changes and oxidative stress [60], as well as maternal stressors that may have long-term consequences on offspring development when affecting the placenta [61].

## 4. Materials and Methods

### 4.1. Animals

Ninety day old female Fisher inbred strain rat dams were kept overnight with males. If the sperm was found in the vaginal smear the next morning, it designated gestation day (GD) 0. On GD 12–13, pregnant dams received 5azaC (5 mg/kg, A 2385, Sigma, St. Louis, MO, USA) dissolved in phosphate buffered saline (PBS) by an i.p. injection. PBN (40 mg/kg, B 7263, Sigma, St. Louis, MO, USA) was injected in the tail vein as pretreatment 1 h before 5azaC or alone. A total of 352 placentas were isolated from four groups of dams i.e., control, PBN, 5azaC, and PBN + 5azaC-treated pregnant dams, 177 at GD 15 and 175 at GD 20.

#### 4.1.1. Sample Isolation and Processing

Pregnant dams were euthanized with i.p. injection of ketamine (0.8 mL/kg, Narketan; Vetoquinol, Bern, Switzerland) and xylazine (0.6 mL/kg, Xylapan; Vetoquinol, Lure. Cedex, France) on GD 15 or GD 20. Placentas were isolated, weighted (all 352), and fixed in St. Marie solution (96% EtOH with 1% glacial acetic acid), dehydrated, and embedded in paraffin. Serial sections (5 µm) were cut for histology or immunohistochemistry on a Leica microtome.

DNA was isolated in TE buffer pH9 with 0.1 mg/mL of Proteinase K and 0.25% of Nonidet P40 (both from (Sigma, St. Louis, MO, USA) at 56 °C for 24 h, heated for 10 min at 95 °C to inactivate Proteinase K, spun and the supernatant was then frozen at −20 °C. DNA concentration and quality were measured with the NanoDrop ND-2000 spectrophotometer (NanoDrop Technologies, Wilmington, NC, USA).

### 4.2. Methods

#### 4.2.1. Bisulfite Conversion and Polymerase Chain Reaction

1000 ng of unpurified isolated genomic DNA was used for bisulfite conversion by EpiTect Plus DNA Bisulfite Kit (#59124; Qiagen, Hilden, Germany), which includes a clean-up step with no necessity for prior purification of DNA. PyroMark PCR Kit (#978703; Qiagen, Hilden, Germany) was used for polymerase chain reaction (PCR) at following conditions: 95 °C for 2 min, 43 °C for 90 s and 72 °C for 60 s for 40 cycles with the PCR forward primer: 5′-GGGTTGGGGATTTAG-3′ and biotinylated reverse primer: 5′ AACCCAAAACCTTA-3′.

#### 4.2.2. Global Methylation Analysis

Global methylation was measured in six samples from different animals per group by pyrosequencing. All steps were conducted as recommended by the manufacturer (Qiagen, Hilden, Germany). Pyromark Q24 Advanced System with PyroMark Q24 CpG Advanced Reagents (#970922; Qiagen, Hilden, Germany) was used for the pyrosequencing reaction; 5′-GGGGATTTAGTTTAGTGGT-3′ was the sequencing primer for the rat ID element [46], while the data obtained were analyzed by the PyroMark Q24 Advanced Software. The global methylation was calculated as the average of the two analyzed CpG’s within the element [47].

#### 4.2.3. Immunohistochemistry

Primary monoclonal antibody, mouse anti-proliferating cell nuclear antigen (PCNA) (1:100, M0879, Dako, Glostrup, Denmark) was used as a proliferation marker, incubated overnight at 4 °C with the negative control reagent (V1617; Dako, Glostrup, Denmark). Next, LSAB2 System-HRP visualization reagent for use on rats (K0609; Dako, Glostrup, Denmark) was applied according to the manufacturer’s instructions. Antibodies to oxidative/nitrosative stress markers were anti-8-OHdG (1:300, sc-66036) and anti-nitrotyrosine (1:100, sc-32757), both from Santa Cruz Biotechnology, Inc. (Dallas, TX, USA). After overnight incubation at 4 °C, sections were treated with Dako REAL EnVision/HRP, Rabbit/Mouse reagent (K5007, Dako, Agilent Technologies Inc., Santa Clara, CA, USA); 3,3′-diaminobenzidine-tetrahydrochloride (DAB) was used for signal staining and hematoxylin for counterstaining. Immunohistochemistry was done on at least 6 samples from different groups of treated animals.

### 4.3. Analysis

#### 4.3.1. Quantitative Stereological Analysis

Binocular light microscope ‘‘Nikon Alphaphot’’ × 400 and the Weibel’s 42-points multipurpose test system (M42) was used for stereological analysis [62]. The length of test lines (Lt) was 1.008 mm and the area tested (At) was 0.0837 mm^2^ for each analyzed microscopic field. The sample size was calculated after analyzing samples on 10 fields with a 95% confidence interval by the formula n = (200/y · s/x)^2^, including the number of fields that have to be analyzed (n), the orientation sample’s arithmetic mean (x), the orientation sample’s standard deviation (SD) (s) and the allowed variation from the arithmetic mean (y) [63]. IHC signal-positive cells were counted in samples and the numerical density (N_v_) was calculated by the formula N_v_ = N/At ·D, including the number of signal-positive cells on the tested area (N), the mean tangential particle diameter measured by 3D Ellipse stereological program for 100 cells (D). D (basal cell nucleus) was 0.015 mm, used for PCNA staining measurement. Oxidative stress analysis was preformed for both basal and labyrinth layer cytoplasmic signal: D (basal cell) 0.032 mm and D (labyrinth layer cell) 0.016 mm. The analysis was done on at least 6 samples from different groups of treated animals.

#### 4.3.2. Apoptotic Index

Ten microscopic fields per sample of hematoxylin–eosin-stained specimens were randomly selected and analyzed as previously described [30,64]. Three placentas from three different dams per group were analyzed.

#### 4.3.3. Statistical Methods/Data Analysis

Statistical evaluation of placental weights, numerical density, apoptotic indices and global DNA methylation levels within groups were evaluated using the ANOVA test with Newman-Keuls multiple comparison post hoc test, or Kruskal–Wallis test with Dunn’s multiple-comparison post hoc test. The coefficient of variation was expressed as SD/mean × 100. Before analyses, the descriptive statistics were performed with the D’Agostino and Pearson test. GraphPad Prism software (version 6.0, GraphPad Software, Inc., San Diego, CA, USA) was used for data analysis. Statistical significance was set at *p* ≤ 0.05.

## Figures and Tables

**Figure 1 ijms-23-00603-f001:**
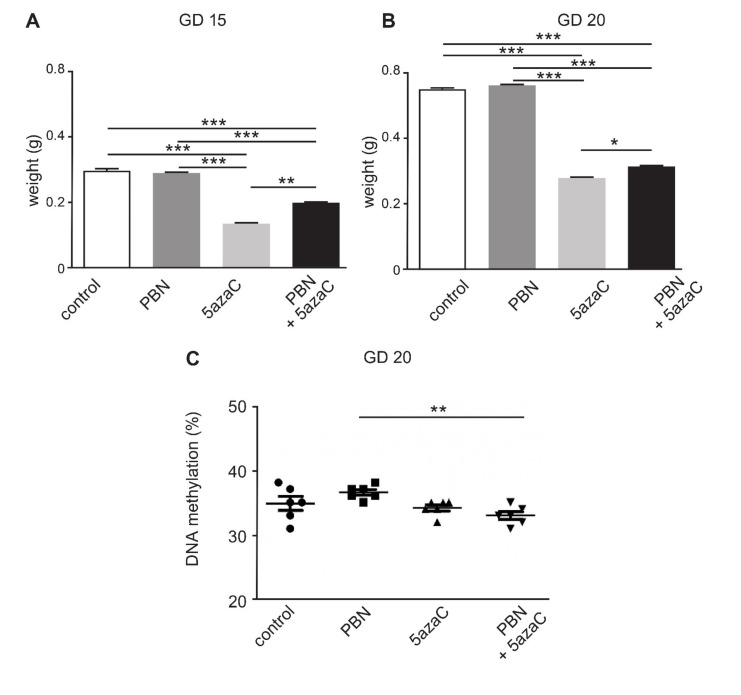
Weight and global DNA methylation of rat placentas from dams treated with N-tert-Butyl-α-phenylnitron (PBN) and/or 5-azacytidine (5azaC) on gestational day (GD) 12–13. Weight of placentas isolated on GD 15 (**A**) or GD 20 (**B**). Percentage of DNA methylation in 20-days old placentas of dams (**C**). ANOVA (**A**,**B**) or Kruskall Wallis test (**C**), * *p* < 0.05, ** *p* < 0.001, *** *p* < 0.0001.

**Figure 2 ijms-23-00603-f002:**
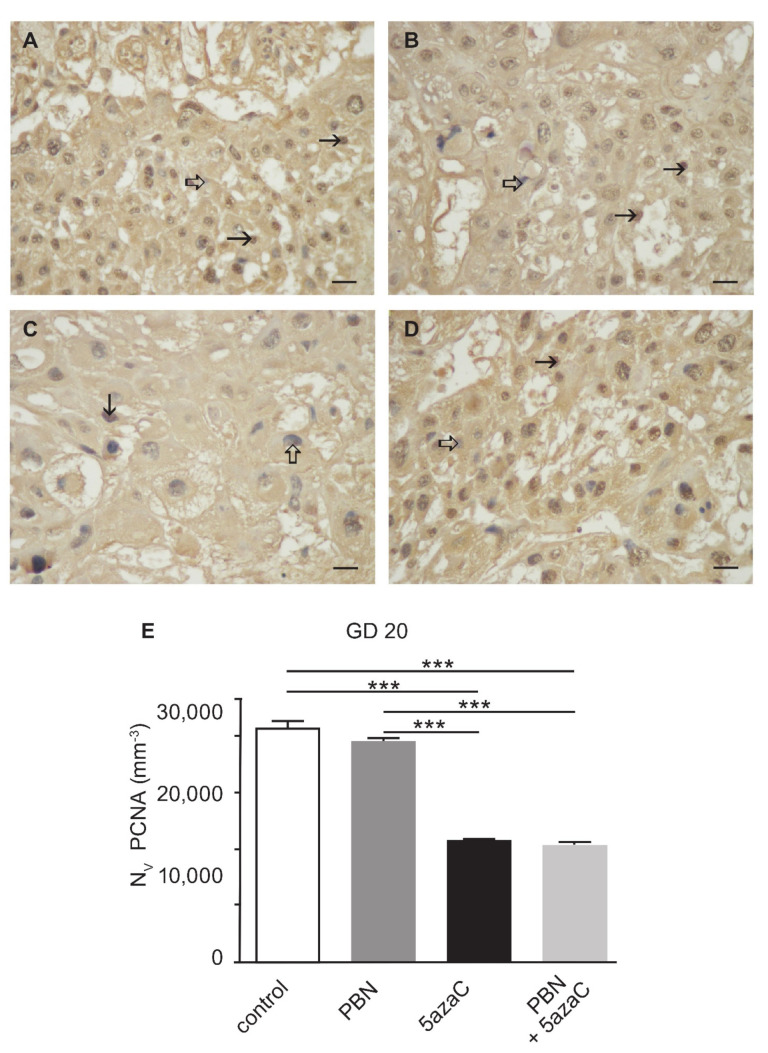
PCNA expression in the GD 20 rat placenta treated with PBN and/or 5-azaC on days GD 12–13. (**A**) control, (**B**) PBN, (**C**) 5azaC, (**D**) PBN + 5azaC. Arrow–PCNA-positive cell, thick arrow–internal negative control. DAB, haematoxylin counterstain. Scale bar 25 µm. (**E**) Stereological quantification of PCNA-positive signals by numerical density (Nv). ANOVA, *** *p* < 0.0001.

**Figure 3 ijms-23-00603-f003:**
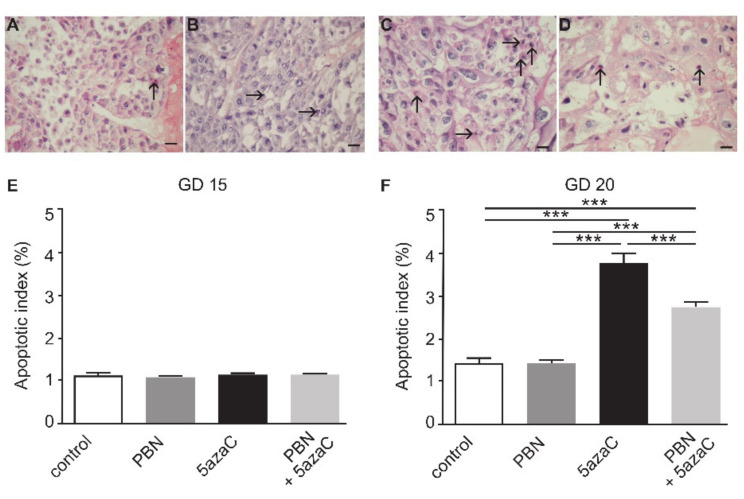
Apoptosis in rat placentas. Representative micrographs of apoptotic cells (arrows) in the basal layer of placentas isolated on GD 15: control (**A**), 5azaC-treatment (**B**). Placentas isolated on the GD 20: control (**C**), 5azaC-treatment (**D**). Hematoxylin and eosin, scale bar 25 µm. Apoptotic index GD 15 (**E**), GD 20 (**F**). ANOVA, *** *p* < 0.0001.

**Figure 4 ijms-23-00603-f004:**
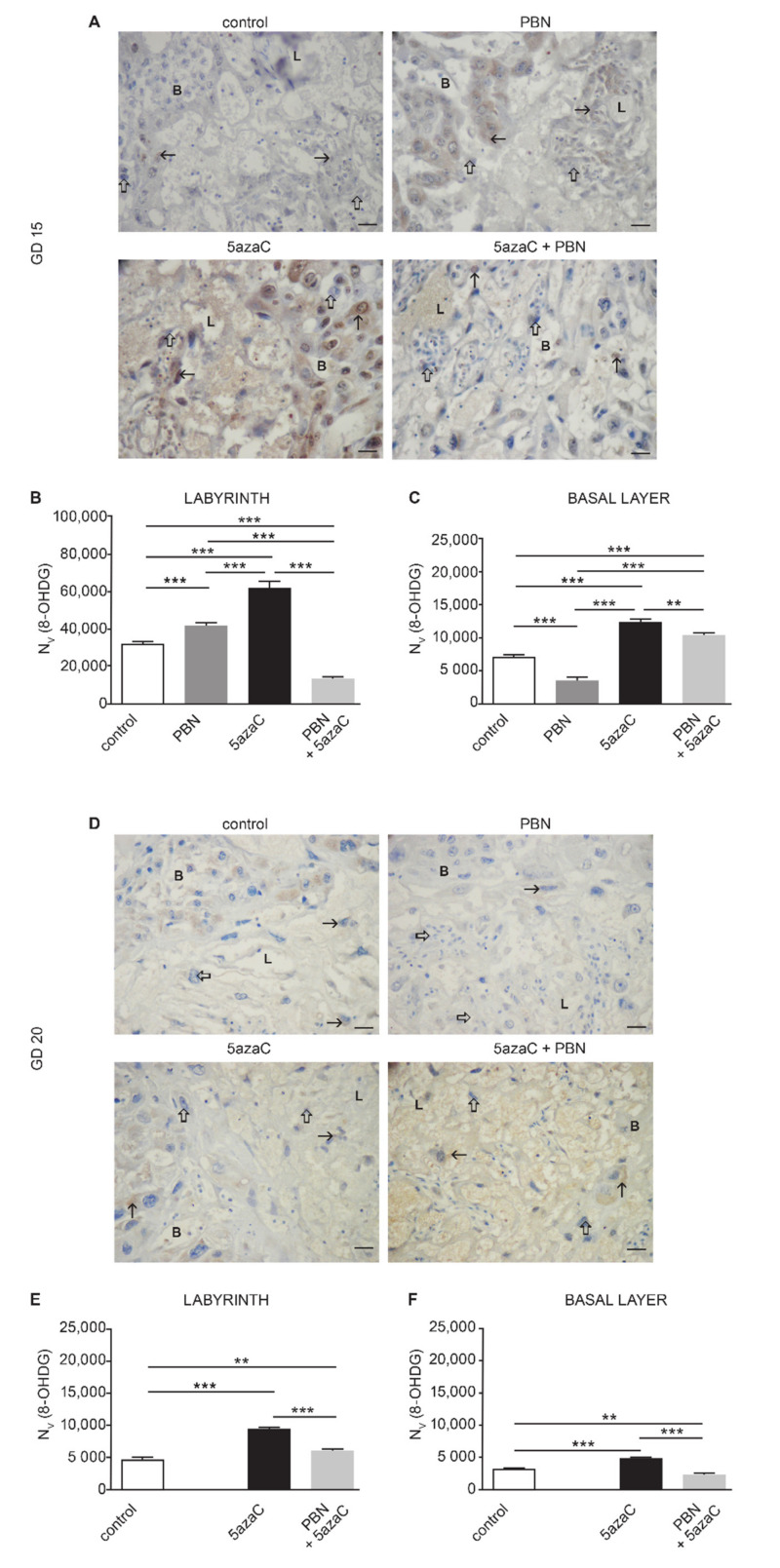
Oxoguanosine (8-OHdG) expression in the rat placenta (labyrinth-L and basal layer-B) treated with PBN and/or 5azaC on GD 12–13 and isolated on GD 15 or GD 20. A, D Images of 8-OHdG expression in placentas isolated on GD 15 (**A**) or GD 20 (**D**). Numerical density (Nv) of 8-OHdG-positive signals (arrow) in the labyrinth of GD 15 (**B**) and GD 20 (**E**) and basal layer of GD 15 (**C**) and GD 20 (**F**). Thick arrow–internal negative control. DAB, haematoxylin counterstain. Scale bar 25 µm. ANOVA, ** *p* < 0.001, *** *p* < 0.0001.

**Figure 5 ijms-23-00603-f005:**
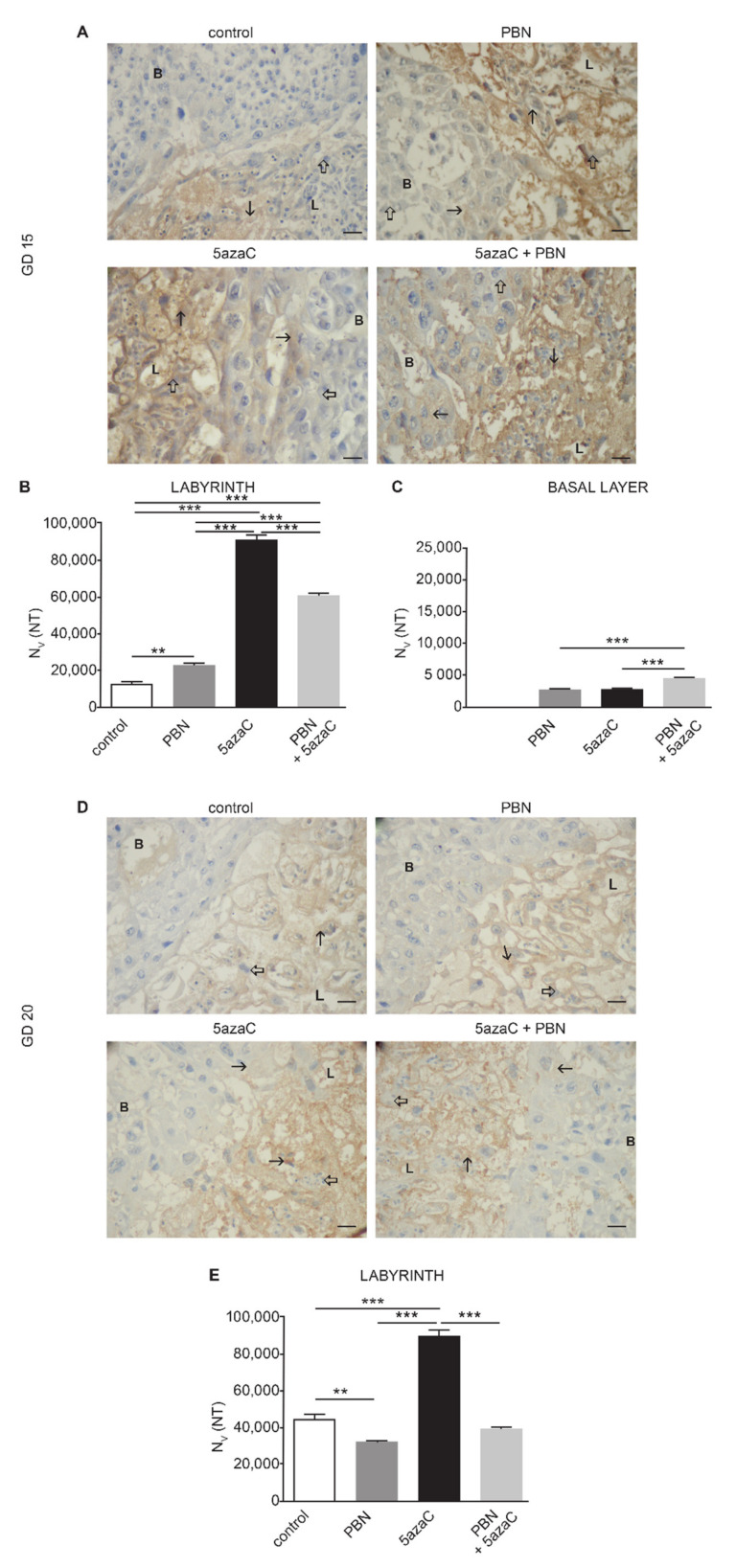
Nitrotyrosine (NT) expression in the placental labyrinth (L) and basal layer (B) treated with PBN and/or 5azaC on GD 12–13 and isolated on the GD 15 or GD 20. A, D Images of nitrotrosine expression (arrow) in placentas isolated on GD 15 (**A**) or GD 20 (**D**). Numerical density (Nv) of NT-positive signals in the labyrinth of GD 15 (**B**) and GD 20 (**E**) and GD 15-basal layer (**C**). Thick arrow–internal negative control. DAB, haematoxylin counterstain. Scale bar 25 µm. ANOVA, ** *p* < 0.001, *** *p* < 0.0001.

## Data Availability

Not applicable.

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
