# Peer review of "Extended Prophylactic Effect of N-tert-Butyl-α-phenylnitron against Oxidative/Nitrosative Damage Caused by the DNA-Hypomethylating Drug 5-Azacytidine in the Rat Placenta"

_ijms, 2022, doi:10.3390/ijms23020603_

Round 1

Reviewer 1 Report

The authors present here an interesting and well-told study regarding protective effects of N-tert-Butyl-α-phenylnitron against ROS/RNS damage induced by 5-azauracile, in continuity with their previously published paper (doi: 10.1089/scd.2018.0194). The work is very well designed and the results clearly presented. Unfortunately, Materials & Methods is mostly copied from the above mentioned paper, please revise this section in order to avoid plagiarism. I have just more notes for the authors:

  • The author state that the study was carried out on a large set of animals, but never explicit this number in the Materials and Method section. Same for the number of samples used in each experiment. 
  • Please resize y-axis' scale of graphs in Figure 4 and 5. Some of them are unreadable.
  • Please correct the "Institutional Review Board Statement": Name of institute, protocol code and date are not reported. 
  • Is there a reason why the authors chose only two biomarkers and one technique (IHC) for assess reduction of oxidative stress induced by PBN? More results obtained from different techniques and/or more biomarkers could have made the conclusions of this work more solid.

Author Response

Reviewer 1

The authors present here an interesting and well-told study regarding protective effects of N-tert-Butyl-α-phenylnitron against ROS/RNS damage induced by 5-azauracile, in continuity with their previously published paper (doi: 10.1089/scd.2018.0194). The work is very well designed and the results clearly presented. Unfortunately, Materials & Methods is mostly copied from the above mentioned paper, please revise this section in order to avoid plagiarism.

The Materials & Methods is now changed as much we possibly could to avoid plagiarism but to keep the significant points of the way the study was conducted. As the studies from the previously published paper and the currently revised one are based on the same techniques and principles, similarities in the description are unavoidable to a certain degree.

 I have just more notes for the authors:

The author state that the study was carried out on a large set of animals, but never explicit this number in the Materials and Method section. Same for the number of samples used in each experiment.

You may find the numbers of animals, placentas and samples per method now in Results and Material and Methods sections.

Please resize y-axis' scale of graphs in Figure 4 and 5. Some of them are unreadable.

The y-axis is now corrected. Please note that no, some of the graph’s y-axis is set to a maximum of 100000, and some to 25000.

Please correct the "Institutional Review Board Statement": Name of institute, protocol code and date are not reported.

This sentence is deleted because in continuation of the text we give the required information.

Is there a reason why the authors chose only two biomarkers and one technique (IHC) for assess reduction of oxidative stress induced by PBN? More results obtained from different techniques and/or more biomarkers could have made the conclusions of this work more solid.

We wanted to show the final effect in the single cells of placental tissue with the quantification of the same biomarkers that we had previously used in:  (ref 30) Sobočan, N. et al. A Free Radical Scavenger Ameliorates Teratogenic Activity of a DNA Hypomethylating Hematological Therapeutic. Stem cells and development 2019, 28, 717-733, doi:10.1089/scd.2018.0194. This allows us to note the histological location of the expressed biomarkers, which, as we could see, differs significantly between the placental layers. Additionally, hydroxynonenal IHC staining was performed on a pilot study, but the signal was too scarce to quantify in all groups.

Reviewer 2 Report

In this work, Sobočan et al. investigated the prophylactic effect of N-tert-Butyl-α-phenylnitron (PBN) on placenta of rats treated with the hypomethylating drug 5-azacytidine (5-azaC), used as a therapy of myelodysplastic syndromes. This article appears as a continuation of a previous work in which the authors assessed the beneficial effects of PBN on the normal development of the offspring of rats treated with 5-azaC (doi:10.1089/scd.2018.0194).

The work is well organized and clearly written although the results can be presented more accurately. Moreover, I believe that the authors should emphasise the possible impact of their investigation even when describing the purpose of the paper. FDA and EMA do not recommend 5-azaC use during pregnancy especially during the first trimester. It would be interesting to evaluate the effects of PBN administration during pregnancy in rats that were treated with 5-azaC prior to pregnancy and that continue (or not continue) 5-azaC treatment during pregnancy. If the authors have not the possibility to perform these experiments, I recommend commenting this aspect in the discussion.

I also have some minor comments:

  • When an acronym has been defined for the first time, it is not necessary to repeat its definition (e.g., PBN in line 52).
  • Line 62 “promoters” instead of “promotors”
  • In the Results section, please carefully report results and the statistical significance of the comparisons among all tested groups both in the text as well in all figures (e.g., in figure 1, the statistical significance of the comparisons with controls and between PBN and PBN+5azaC groups, is missing). Now, statistical significance of the results in the figure is not clearly and correctly reported. Please also include reference to the figure panels to help the reader to observe the described results in the figures.
  • The demethylating effect of 5azaC seems quite modest. Please provide in the text percentage of methylation levels in the different analysed groups and modify y-axis scale to allow visualization of the differences in DNA methylation levels among groups. Moreover, please indicate in the Results and comment in the Discussion that there is not any statistically significant difference between 5azaC treated rats with those receiving prior PBN.
  • Explanation of panel C of Figure 1 is missing.
  • Please avoid commenting on the results in the Results section. All provided explanations of results should be moved to the Discussion section.
  • I could not visualize the entire Figure 4 and 5 (in particular, panels C and F of Figure 4 and panel C of Figure 5, that are cut). Moreover, please carefully check this sentence “In the basal layer of GD 20 placentas, oxoguanosine was absent from both the labyrinth and the basal layer of PBN-treated dams.”. I assume that is referred to the absence of oxoguanosine at GD20 in both the labyrinth and the basal layer of PBN-treated dams.
  • Please avoid “cause-effects” sentences (such as the one at line 216-218) and describe and discuss results as correlations/associations. Please comment the fact that PBN alone was correlated with significantly different levels of oxidative stress markers compared with controls and the effects of PBN on the expression of these markers was different at GD15 and GD20 (e.g., increased expression levels of Oxoguanosine at GD15 and undetectable expression at GD20).
  • Referring to this sentence at line 248: “Regarding cell proliferation, neither 5azaC nor PBN prophylaxis influenced cell proliferation in the GD 20 placentas”, figure 2 clearly showed a statistically significant decrease of PCNA expression in 5azaC treated rats that is present regardless PBM pre-treatment.
  • The authors should clearly describe the strategy to estimate global DNA methylation level based on the analysis of short interspersed repetitive elements in the rat genome.
  • I found interesting that the authors did not purify the genomic DNA prior bisulfite conversion. Could you please motivate this choice in the rebuttal letter?

Author Response

Reviewer 2

In this work, Sobočan et al. investigated the prophylactic effect of N-tert-Butyl-α-phenylnitron (PBN) on placenta of rats treated with the hypomethylating drug 5-azacytidine (5-azaC), used as a therapy of myelodysplastic syndromes. This article appears as a continuation of a previous work in which the authors assessed the beneficial effects of PBN on the normal development of the offspring of rats treated with 5-azaC (doi:10.1089/scd.2018.0194).

The work is well organized and clearly written although the results can be presented more accurately. Moreover, I believe that the authors should emphasise the possible impact of their investigation even when describing the purpose of the paper. FDA and EMA do not recommend 5-azaC use during pregnancy especially during the first trimester. It would be interesting to evaluate the effects of PBN administration during pregnancy in rats that were treated with 5-azaC prior to pregnancy and that continue (or not continue) 5-azaC treatment during pregnancy. If the authors have not the possibility to perform these experiments, I recommend commenting this aspect in the discussion.

Our previous work on rats showed the teratogenicity of 5azaC and its negative effect on placenta.  lines 83-84 : "... and our previous results showed that 5azaC impaired placental development and changed its glycoprotein composition [21-23].

Now we wanted to start to reveal the possible mechanisms for such an activity specifically in the placenta. Treating females with 5azaC before pregnancy would make the study very complicated given that  first we would have to investigate its effect on the whole adult female organism that is certainly not so susceptible to 5azaC activity as are the developing embryo and placenta. There is a huge difference between the mother and the embryo in reactivity to external agents, and especially epigenetic ones that have the major role in development.  Lines 68-71 "During placental development, DNA methylation changes may be detrimental to the placenta [21-23] because DNA methylation is involved in the regulation of gene activity necessary for differentiation and other developmental processes [24-26]. "  

However, we now put in the discussion:

"Ogbodo et al. proposed testing of free radicals and antioxidant status before human pregnancy and in early-pregnancy (Ogbodo S. O., Okaka A. N. C., Nwagha U. I. Ejezie F. E. Free Radicals and Antioxidants Status in Pregnancy: Need for Pre- and Early Pregnancy Assessment American Journal of Medicine and Medical Sciences 2014, 4(6): 230-235 DOI: 10.5923/j.ajmms.20140406.06.). It is possible that a future basic reserch using treatments such as ours before the pregnancy and at earlier stages than done in this and our previous research (30) may further support that proposition. "

I also have some minor comments:

  • When an acronym has been defined for the first time, it is not necessary to repeat its definition (e.g., PBN in line 52).
  • Changed: "The spin-trap PBN [10] protects mammalian embryos......"
  • Line 62 “promoters” instead of “promotors”
  • Changed: "...promoters of tumor supressor genes"

  • In the Results section, please carefully report results and the statistical significance of the comparisons among all tested groups both in the text as well in all figures (e.g., in figure 1, the statistical significance of the comparisons with controls and between PBN and PBN+5azaC groups, is missing). Now, statistical significance of the results in the figure is not clearly and correctly reported. Please also include reference to the figure panels to help the reader to observe the described results in the figures.
  • The P values and references to the figures are now added to the text, for better clarity. We have rewritten the results accordingly. 

  • The demethylating effect of 5azaC seems quite modest. Please provide in the text percentage of methylation levels in the different analysed groups and modify y-axis scale to allow visualization of the differences in DNA methylation levels among groups. Moreover, please indicate in the Results and comment in the Discussion that there is not any statistically significant difference between 5azaC treated rats with those receiving prior PBN.
  • Indicated in the results and commented in discussion.
  • The percentage of methylation levels are added and the y-axis scale is modified from 0 – 100 to 20 – 50.

  • Explanation of panel C of Figure 1 is missing.
  • There is an explanation in legend but (C) was missing
  • " Percentage of DNA methylation in 20-days old placentas of dams (C). ANOVA (A, B) or Kruskall Wallis test (C)"

  • Please avoid commenting on the results in the Results section. All provided explanations of results should be moved to the Discussion section.
    • Certain sentences with a discussion – like conclusions were moved to the appropriate section. Some serve only to summarize the result, section by section.

  • I could not visualize the entire Figure 4 and 5 (in particular, panels C and F of Figure 4 and panel C of Figure 5, that are cut). Moreover, please carefully check this sentence “In the basal layer of GD 20 placentas, oxoguanosine was absent from both the labyrinth and the basal layer of PBN-treated dams.”. I assume that is referred to the absence of oxoguanosine at GD20 in both the labyrinth and the basal layer of PBN-treated dams.
    • Now all Figures are visible.
    • Changed: "In GD 20 placentas, oxoguanosine was absent from both the labyrinth and the basal layer of PBN-treated dams."

  • Please avoid “cause-effects” sentences (such as the one at line 216-218) and describe and discuss results as correlations/associations. Please comment the fact that PBN alone was correlated with significantly different levels of oxidative stress markers compared with controls and the effects of PBN on the expression of these markers was different at GD15 and GD20 (e.g., increased expression levels of Oxoguanosine at GD15 and undetectable expression at GD20).
  • Changed: "Our results obtained on inbred animals in comparison to untreated controls have shown that a DNA hypomethylating agent is associated to elevated levels of ROS markers, specifically 8-hydroxy-2'-deoxyguanosine (8-OHdG) and nitrotyrosine in the placenta."

  • Discussed: " Our experimental results have shown that prophylactic in vivo pretreatment of the pregnant rat dams with the antioxidant PBN generally alleviated induction of ROS/RNS by a DNA hypomethylating drug in the mammalian placenta. The only exception was in the basal layer of GD 15 placentas where pretreatment caused significantly higher level of nitrotyrosine than in 5azaC-treated samples. The treatment with PBN only was mostly associated with lower levels of ROS/RNS markers than in controls. However, in the GD 15 labyrinth PBN-treatment alone was correlated with significantly higher levels of both oxoguanosine and nitrotyrosine than in controls. All of above unexpected findings are possibly associated with other PBN activities than antioxidative, such as the activity of a nitrone on gene expression [35,36]. We also found that at GD 15 nitrotyrosine in the basal layer of the placenta was of a low, but detectable level in all treated except in controls, while at GD20 it was always too low for quantification. It was described before that nitrotyrosine residues were present in the placenta in association with altered placental function caused by maternal diseases [37] that obviously influenced mothers more than our dams treated with 5azaC.”

  • Referring to this sentence at line 248: “Regarding cell proliferation, neither 5azaC nor PBN prophylaxis influenced cell proliferation in the GD 20 placentas”, figure 2 clearly showed a statistically significant decrease of PCNA expression in 5azaC treated rats that is present regardless PBM pre-treatment.
  • Changed: "Regarding cell proliferation, 5azaC diminished it, and PBN prophylaxis had no influence in the GD 20 placentas."

  • The authors should clearly describe the strategy to estimate global DNA methylation level based on the analysis of short interspersed repetitive elements in the rat genome.
  • "In our global DNA methylation research we used the rat B1 ID element/SINE [46]. Such SINEs are appropriate for the analysis of global DNA methylation in the rat because they are present in over one hundred thousand copies per haploid genome. Global methylation was calculated as the average of the two analyzed CpG’s within the element [47,48].

  • I found interesting that the authors did not purify the genomic DNA prior bisulfite conversion. Could you please motivate this choice in the rebuttal letter?
  • "1000 ng of unpurified isolated genomic DNA was used for bisulfite conversion by EpiTect Plus DNA Bisulfite Kit (#59124; Qiagen) that includes a clean-up step with no necessity for prior purification of DNA."

EpiTect Plus Bisulfite Kits (qiagen.com)

Round 2

Reviewer 1 Report

I'd like to thank the authors to have properly answered to all my comments. 

Author Response

The authors, too, would like to thank the reviewer for forming a better manuscript together.

Reviewer 2 Report

The authors have addressed most of my concerns. I only recommend checking again the reported statistically significance in the figures (e.g. statistical significance between 5azaC and control is missing in Figure 1B, as well as statistical significance between PBM treated dams and dams treated with 5azaC + PBN in Figure 1A).

Author Response

We have now reviewed our results in detail and added all the data on the statistical significance in all the images. We thank the editor for the improvement of our manuscript.
